# Fast and Accurate Inference of Plackett–Luce Models

**Lucas Maystre**
EPFL
lucas.maystre@epfl.ch

**Matthias Grossglauser**
EPFL
matthias.grossglauser@epfl.ch

## Abstract

We show that the maximum-likelihood (ML) estimate of models derived from Luce's choice axiom (e.g., the Plackett–Luce model) can be expressed as the stationary distribution of a Markov chain. This conveys insight into several recently proposed spectral inference algorithms. We take advantage of this perspective and formulate a new spectral algorithm that is significantly more accurate than previous ones for the Plackett–Luce model. With a simple adaptation, this algorithm can be used iteratively, producing a sequence of estimates that converges to the ML estimate. The ML version runs faster than competing approaches on a benchmark of five datasets. Our algorithms are easy to implement, making them relevant for practitioners at large.

## 1 Introduction

Aggregating pairwise comparisons and partial rankings are important problems with applications in econometrics [1], psychometrics [2, 3], sports ranking [4, 5] and multiclass classification [6]. One possible approach to tackle these problems is to postulate a statistical model of discrete choice. In this spirit, Luce [7] stated the *axiom of choice* in a foundational work published over fifty years ago. Denote $p(i \mid A)$ the probability of choosing item $i$ when faced with alternatives in the set $A$. Given two items $i$ and $j$, and any two sets of alternatives $A$ and $B$ containing $i$ and $j$, the axiom posits that

$$\frac{p(i \mid A)}{p(j \mid A)} = \frac{p(i \mid B)}{p(j \mid B)}. \tag{1}$$

In other words, the odds of choosing item $i$ over item $j$ are independent of the rest of the alternatives. This simple assumption directly leads to a unique parametric choice model, known as the Bradley–Terry model in the case of pairwise comparisons, and the Plackett–Luce model in the generalized case of $k$-way rankings. In this paper, we highlight a connection between the maximum-likelihood (ML) estimate under these models and the stationary distribution of a Markov chain parameterized by the observed choices. Markov chains were already used in recent work [8, 9, 10] to aggregate pairwise comparisons and rankings. These approaches reduce the problem to that of finding a stationary distribution. By formalizing the link between the likelihood of observations under the choice model and a certain Markov chain, we unify these algorithms and explicate them from an ML inference perspective. We will also take a detour, and use this link in the reverse direction to give an alternative proof to a recent result on the error rate of the ML estimate [11], by using spectral analysis techniques.

Beyond this, we make two contributions to statistical inference for this model. First, we develop a simple, consistent and computationally efficient spectral algorithm that is applicable to a wide range of models derived from the choice axiom. The exact formulation of the Markov chain used in the algorithm is distinct from related work [9, 10] and achieves a significantly better statistical efficiency at no additional computational cost. Second, we observe that with a small adjustment, the algorithm can be used iteratively, and it then converges to the ML estimate. An evaluation on five real-world datasets reveals that it runs consistently faster than competing approaches and has a much more predictable performance that does not depend on the structure of the data. The key step, finding a stationary distribution, can be offloaded to commonly available linear-algebra primitives, which

makes our algorithms scale well. Our algorithms are intuitively pleasing, simple to understand and implement, and they outperform the state of the art, hence we believe that they will be highly useful to practitioners.

The rest of the paper is organized as follows. We begin by introducing some notations and presenting a few useful facts about the choice model and about Markov chains. By necessity, our exposition is succinct, and the reader is encouraged to consult Luce [7] and Levin et al. [12] for a more thorough exposition. In Section 2, we discuss related work. In Section 3, we present our algorithms, and in Section 4 we evaluate them on synthetic and real-world data. We conclude in Section 5.

**Discrete choice model.** Denote by $n$ the number of items. Luce's axiom of choice implies that each item $i \in \{1, \ldots, n\}$ can be parameterized by a positive strength $\pi_i \in \mathbf{R}_{>0}$ such that $p(i \mid A) = \pi_i / \sum_{j \in A} \pi_j$ for any $A$ containing $i$. The strengths $\boldsymbol{\pi} = [\pi_i]$ are defined up to a multiplicative factor; for identifiability, we let $\sum_i \pi_i = 1$. An alternative parameterization of the model is given by $\theta_i = \log(\pi_i)$, in which case the model is sometimes referred to as *conditional logit* [1].

**Markov chain theory.** We represent a finite, stationary, continuous-time Markov chain by a directed graph $G = (V, E)$, where $V$ is the set of states and $E$ is the set of transitions with positive rate. If $G$ is strongly connected, the Markov chain is said to be ergodic and admits a unique *stationary distribution* $\boldsymbol{\pi}$. The global balance equations relate the transition rates $\lambda_{ij}$ to the stationary distribution as follows:

$$\sum_{j \neq i} \pi_i \lambda_{ij} = \sum_{j \neq i} \pi_j \lambda_{ji} \quad \forall i. \tag{2}$$

The stationary distribution is therefore invariant to changes in the time scale, i.e., to a rescaling of the transition rates. In the supplementary file, we briefly discuss how to find $\boldsymbol{\pi}$ given $[\lambda_{ij}]$.

## 2    Related work

Spectral methods applied to ranking and scoring items from noisy choices have a long-standing history. To the best of our knowledge, Saaty [13] is the first to suggest using the leading eigenvector of a matrix of inconsistent pairwise judgments to score alternatives. Two decades later, Page et al. [14] developed PageRank, an algorithm that ranks Web pages according to the stationary distribution of a random walk on the hyperlink graph. In the same vein, Dwork et al. [8] proposed several variants of Markov chains for aggregating heterogeneous rankings. The idea is to construct a random walk that is biased towards high-ranked items, and use the ranking induced by the stationary distribution. More recently, Negahban et al. [9] presented Rank Centrality, an algorithm for aggregating pairwise comparisons close in spirit to that of [8]. When the data is generated under the Bradley–Terry model, this algorithm asymptotically recovers model parameters with only $\omega(n \log n)$ pairwise comparisons. For the more general case of rankings under the Plackett–Luce model, Azari Soufiani et al. [10] propose to break rankings into pairwise comparisons and to apply an algorithm similar to Rank Centrality. They show that the resulting estimator is statistically consistent. Interestingly, many of these spectral algorithms can be related to the method of moments, a broadly applicable alternative to maximum-likelihood estimation.

The history of algorithms for maximum-likelihood inference under Luce's model goes back even further. In the special case of pairwise comparisons, the same iterative algorithm was independently discovered by Zermelo [15], Ford [16] and Dykstra [17]. Much later, this algorithm was explained by Hunter [18] as an instance of minorization-maximization (MM) algorithm, and extended to the more general choice model. Today, Hunter's MM algorithm is the *de facto* standard for ML inference in Luce's model. As the likelihood is concave, off-the-shelf optimization procedures such as the Newton-Raphson method can also be used, although they have been been reported to be slower and less practical [18]. Recently, Kumar et al. [19] looked at the problem of finding the transition matrix of a Markov chain, given its stationary distribution. The problem of inferring Luce's model parameters from data can be reformulated in their framework, and the ML estimate is the solution to the inversion of the stationary distribution. Their work stands out as the first to link ML inference to Markov chains, albeit very differently from the way presented in our paper. Beyond algorithms, properties of the maximum-likelihood estimator in this model were studied extensively. Hajek et al. [11] consider the Plackett–Luce model for $k$-way rankings. They give an upper bound to the estimation error and show that the ML estimator is minimax-optimal. In summary, they show that only $\omega(n/k \log n)$ samples are enough to drive the mean-square error down to zero, as $n$ increases. Rajkumar and Agarwal [20]

consider the Bradley–Terry model for pairwise comparisons. They show that the ML estimator is able to recover the correct ranking, even when the data is generated as per another model, e.g., Thurstone's [2], as long as a so-called *low-noise* condition is satisfied. We also mention that as an alternative to likelihood maximization, Bayesian inference has also been proposed. Caron and Doucet [21] present a Gibbs sampler, and Guiver and Snelson [22] propose an approximate inference algorithm based on expectation propagation.

In this work, we provide a unifying perspective on recent advances in spectral algorithms [9, 10] from a maximum-likelihood estimation perspective. It turns out that this perspective enables us to make contributions on both sides: On the one hand, we develop an improved and more general spectral ranking algorithm, and on the other hand, we propose a faster procedure for ML inference by using this algorithm iteratively.

## 3 Algorithms

We begin by expressing the ML estimate under the choice model as the stationary distribution of a Markov chain. We then take advantage of this formulation to propose novel algorithms for model inference. Although our derivation is made in the general choice model, we will also discuss implications for the special cases of pairwise data in Section 3.3 and $k$-way ranking data in Section 3.4. Suppose that we collect $d$ independent observations in the multiset $\mathcal{D} = \{(c_\ell, A_\ell) \mid \ell = 1, \ldots, d\}$. Each observation consists of a choice $c_\ell$ among a set of alternatives $A_\ell$; we say that $i$ *wins over* $j$ and denote by $i \succ j$ whenever $i, j \in A$ and $c_\ell = i$. We define the directed *comparison graph* as $G_\mathcal{D} = (V, E)$, with $V = \{1, \ldots, n\}$ and $(j, i) \in E$ if $i$ wins at least once over $j$ in $\mathcal{D}$. In order to ensure that the ML estimate is well-defined, we make the standard assumption that $G_\mathcal{D}$ is strongly connected [16, 18]. In practice, if this assumption does not hold, we can consider each strongly connected component separately.

### 3.1 ML estimate as a stationary distribution

For simplicity, we denote the model parameter associated with item $c_\ell$ by $\pi_\ell$. The log-likelihood of parameters $\boldsymbol{\pi}$ given observations $\mathcal{D}$ is

$$\log \mathcal{L}(\boldsymbol{\pi} \mid \mathcal{D}) = \sum_{\ell=1}^{d} \left( \log \pi_\ell - \log \sum_{j \in A_\ell} \pi_j \right). \tag{3}$$

For each item, we define two sets of indices. Let $W_i \doteq \{\ell \mid i \in A_\ell, c_\ell = i\}$ and $L_i \doteq \{\ell \mid i \in A_\ell, c_\ell \neq i\}$ be the indices of the observations where item $i$ wins over and loses against the alternatives, respectively. The log-likelihood function is strictly concave and the model admits a unique ML estimate $\hat{\boldsymbol{\pi}}$. The optimality condition $\nabla_{\hat{\boldsymbol{\pi}}} \log \mathcal{L} = 0$ implies

$$\frac{\partial \log \mathcal{L}}{\partial \hat{\pi}_i} = \sum_{\ell \in W_i} \left( \frac{1}{\hat{\pi}_i} - \frac{1}{\sum_{j \in A_\ell} \hat{\pi}_j} \right) - \sum_{\ell \in L_i} \frac{1}{\sum_{j \in A_\ell} \hat{\pi}_j} = 0 \quad \forall i \tag{4}$$

$$\iff \sum_{j \neq i} \left( \sum_{\ell \in W_i \cap L_j} \frac{\hat{\pi}_j}{\sum_{t \in A_\ell} \hat{\pi}_t} - \sum_{\ell \in W_j \cap L_i} \frac{\hat{\pi}_i}{\sum_{t \in A_\ell} \hat{\pi}_t} \right) = 0 \quad \forall i. \tag{5}$$

In order to go from (4) to (5), we multiply by $\hat{\pi}_i$ and rearrange the terms. To simplify the notation, let us further introduce the function

$$f(\mathcal{S}, \boldsymbol{\pi}) \doteq \sum_{A \in \mathcal{S}} \frac{1}{\sum_{i \in A} \pi_i}, \tag{6}$$

which takes observations $\mathcal{S} \subseteq \mathcal{D}$ and an instance of model parameters $\boldsymbol{\pi}$, and returns a non-negative real number. Let $\mathcal{D}_{i \succ j} \doteq \{(c_\ell, A_\ell) \in \mathcal{D} \mid \ell \in W_i \cap L_j\}$, i.e., the set of observations where $i$ wins over $j$. Then (5) can be rewritten as

$$\sum_{j \neq i} \hat{\pi}_i \cdot f(\mathcal{D}_{j \succ i}, \hat{\boldsymbol{\pi}}) = \sum_{j \neq i} \hat{\pi}_j \cdot f(\mathcal{D}_{i \succ j}, \hat{\boldsymbol{\pi}}) \quad \forall i. \tag{7}$$

| **Algorithm 1** Luce Spectral Ranking | **Algorithm 2** Iterative Luce Spectral Ranking |
|---|---|
| **Require:** observations $\mathcal{D}$ | **Require:** observations $\mathcal{D}$ |
| 1: $\lambda \leftarrow 0_{n \times n}$ | 1: $\boldsymbol{\pi} \leftarrow [1/n, \ldots, 1/n]^\intercal$ |
| 2: **for** $(i, A) \in \mathcal{D}$ **do** | 2: **repeat** |
| 3:     **for** $j \in A \setminus \{i\}$ **do** | 3:     $\lambda \leftarrow 0_{n \times n}$ |
| 4:       $\lambda_{ji} \leftarrow \lambda_{ji} + n/\|A\|$ | 4:     **for** $(i, A) \in \mathcal{D}$ **do** |
| 5:     **end for** | 5:       **for** $j \in A \setminus \{i\}$ **do** |
| 6: **end for** | 6:         $\lambda_{ji} \leftarrow \lambda_{ji} + 1/\sum_{t \in A} \pi_t$ |
| 7: $\bar{\boldsymbol{\pi}} \leftarrow$ stat. dist. of Markov chain $\lambda$ | 7:       **end for** |
| 8: **return** $\bar{\boldsymbol{\pi}}$ | 8:     **end for** |
| | 9:     $\boldsymbol{\pi} \leftarrow$ stat. dist. of Markov chain $\lambda$ |
| | 10: **until** convergence |

This formulation conveys a new viewpoint on the ML estimate. It is easy to recognize the global balance equations (2) of a Markov chain on $n$ states (representing the items), with transition rates $\lambda_{ji} = f(\mathcal{D}_{i \succ j}, \hat{\boldsymbol{\pi}})$ and stationary distribution $\hat{\boldsymbol{\pi}}$. These transition rates have an interesting interpretation: $f(\mathcal{D}_{i \succ j}, \boldsymbol{\pi})$ is the count of how many times $i$ wins over $j$, weighted by the strength of the alternatives. At this point, it is useful to observe that for any parameters $\boldsymbol{\pi}$, $f(\mathcal{D}_{i \succ j}, \boldsymbol{\pi}) \geq 1$ if $(j, i) \in E$, and 0 otherwise. Combined with the assumption that $G_\mathcal{D}$ is strongly connected, it follows that any $\boldsymbol{\pi}$ parameterizes the transition rates of an ergodic (homogeneous) Markov chain. The ergodicity of the inhomogeneous Markov chain, where the transition rates are constantly updated to reflect the current distribution over states, is shown by the following theorem.

**Theorem 1.** *The Markov chain with inhomogeneous transition rates $\lambda_{ji} = f(\mathcal{D}_{i \succ j}, \boldsymbol{\pi})$ converges to the maximum-likelihood estimate $\hat{\boldsymbol{\pi}}$, for any initial distribution in the open probability simplex.*

*Proof (sketch).* By (7), $\hat{\boldsymbol{\pi}}$ is the unique invariant distribution of the Markov chain. In the supplementary file, we look at an equivalent uniformized discrete-time chain. Using the contraction mapping principle, one can show that this chain converges to the invariant distribution. $\square$

### 3.2 Approximate and exact ML inference

We approximate the Markov chain described in (7) by considering a priori that all alternatives have equal strength. That is, we set the transition rates $\lambda_{ji} \doteq f(\mathcal{D}_{i \succ j}, \boldsymbol{\pi})$ by fixing $\boldsymbol{\pi}$ to $[1/n, \ldots, 1/n]^\intercal$. For $i \neq j$, the contribution of $i$ winning over $j$ to the rate of transition $\lambda_{ji}$ is $n/\|A\|$. In other words, for each observation, the winning item is rewarded by a fixed amount of incoming rate that is evenly split across the alternatives (the chunk allocated to itself is discarded.) We interpret the stationary distribution $\bar{\boldsymbol{\pi}}$ as an estimate of model parameters. Algorithm 1 summarizes this procedure, called *Luce Spectral Ranking* (LSR.) If we consider a growing number of observations, LSR converges to the true model parameters $\boldsymbol{\pi}^*$, even in the restrictive case where the sets of alternatives are fixed.

**Theorem 2.** *Let $\mathcal{A} = \{A_\ell\}$ be a collection of sets of alternatives such that for any partition of $\mathcal{A}$ into two non-empty sets $S$ and $T$, $(\cup_{A \in S} A) \cap (\cup_{A \in T} A) \neq \varnothing$[1]. Let $d_\ell$ be the number of choices observed over alternatives $A_\ell$. Then $\bar{\boldsymbol{\pi}} \to \boldsymbol{\pi}^*$ as $d_\ell \to \infty \; \forall \ell$.*

*Proof (sketch).* The condition on $\mathcal{A}$ ensures that asymptotically $G_\mathcal{D}$ is strongly connected. Let $d \to \infty$ be a shorthand for $d_\ell \to \infty \; \forall \ell$. We can show that if items $i$ and $j$ are compared in at least one set of alternatives, the ratio of transition rates satisfies $\lim_{d \to \infty} \lambda_{ij}/\lambda_{ji} = \pi_j^*/\pi_i^*$. It follows that in the limit of $d \to \infty$, the stationary distribution is $\boldsymbol{\pi}^*$. A rigorous proof is given in the supplementary file. $\square$

Starting from the LSR estimate, we can iteratively refine the transition rates of the Markov chain and obtain a sequence of estimates. By (7), the only fixed point of this iteration is the ML estimate $\hat{\boldsymbol{\pi}}$. We call this procedure I-LSR and describe it in Algorithm 2.

LSR (or one iteration of I-LSR) entails (*a*) filling a matrix of (weighted) pairwise counts and (*b*) finding a stationary distribution. Let $D \doteq \sum_\ell \|A_\ell\|$, and let $S$ be the running time of finding the stationary distribution. Then LSR has running time $O(D + S)$. As a comparison, one iteration of

the MM algorithm [18] is $O(D)$. Finding the stationary distribution can be implemented in different ways. For example, in a sparse regime where $D \ll n^2$, the stationary distribution can be found with the power method in a few $O(D)$ sparse matrix multiplications. In the supplementary file, we give more details about possible implementations. In practice, whether $D$ or $S$ turns out to be dominant in the running time is not a foregone conclusion.

## 3.3 Aggregating pairwise comparisons

A widely-used special case of Luce's choice model occurs when all sets of alternatives contain exactly two items, i.e., when the data consists of pairwise comparisons. This model was proposed by Zermelo [15], and later by Bradley and Terry [3]. As the stationary distribution is invariant to changes in the time-scale, we can rescale the transition rates and set $\lambda_{ji} \doteq |\mathcal{D}_{i \succ j}|$ when using LSR on pairwise data. Let $S$ be the set containing the pairs of items that have been compared at least once. In the case where each pair $(i, j) \in S$ has been compared exactly $p$ times, LSR is strictly equivalent to a continuous-time Markov-chain formulation of Rank Centrality [9]. In fact, our derivation justifies Rank Centrality as an approximate ML inference algorithm for the Bradley–Terry model. Furthermore, we provide a principled extension of Rank Centrality to the case where the number of comparisons observed is unbalanced. Rank Centrality considers transition rates proportional to the *ratio* of wins, whereas (7) justifies making transition rates proportional to the *count* of wins.

Negahban et al. [9] also provide an upper bound on the error rate of Rank Centrality, which essentially shows that it is minimax-optimal. Because the two estimators are equivalent in the setting of balanced pairwise comparisons, the bound also applies to LSR. More interestingly, the expression of the ML estimate as a stationary distribution enables us to reuse the same analytical techniques to bound the error of the ML estimate. In the supplementary file, we therefore provide an alternative proof of the recent result of Hajek et al. [11] on the minimax-optimality of the ML estimate.

## 3.4 Aggregating partial rankings

Another case of interest is when observations do not consist of only a single choice, but of a ranking over the alternatives. We now suppose $m$ observations consisting of $k$-way rankings, $2 \le k \le n$. For conciseness, we suppose that $k$ is the same for all observations. Let one such observation be $\sigma(1) \succ \ldots \succ \sigma(k)$, where $\sigma(p)$ is the item with $p$-th rank. Luce [7] and later Plackett [4] independently proposed a model of rankings where

$$\mathbf{P}\left(\sigma(1) \succ \ldots \succ \sigma(k)\right) = \prod_{r=1}^{k} \frac{\pi_{\sigma(r)}}{\sum_{p=r}^{k} \pi_{\sigma(p)}}. \tag{8}$$

In this model, a ranking can be interpreted as a sequence of $k - 1$ independent choices: Choose the first item, then choose the second among the remaining alternatives, etc. With this point of view in mind, LSR and I-LSR can easily accommodate data consisting of $k$-way rankings, by decomposing the $m$ observations into $d = m(k - 1)$ choices.

Azari Soufiani et al. [10] provide a class of consistent estimators for the Plackett–Luce model, using the idea of breaking rankings into pairwise comparisons. Although they explain their algorithms from a generalized-method-of-moments perspective, it is straightforward to reinterpret their estimators as stationary distributions of particular Markov chains. In fact, for $k = 2$, their algorithm GMM-F is identical to LSR. When $k > 2$ however, breaking a ranking into $\binom{k}{2}$ pairwise comparisons implicitly makes the (incorrect) assumption that these comparisons are statistically independent. The Markov chain that LSR builds breaks rankings into pairwise rate contributions, but weights the contributions differently depending on the rank of the winning item. In Section 4, we show that this weighting turns out to be crucial. Our approach yields a significant improvement in statistical efficiency, yet keeps the same attractive computational cost and ease of use.

## 3.5 Applicability to other models

Several other variants and extensions of Luce's choice model have been proposed. For example, Rao and Kupper [23] extend the Bradley–Terry model to the case where a comparison between two items can result in a tie. In the supplementary file, we show that the ML estimate in the Rao–Kupper model can also be formulated as a stationary distribution, and we provide corresponding adaptations of LSR

and I-LSR. We believe that our algorithms can be generalized to further models that are based on the choice axiom. However, this axiom is key, and other choice models (such as Thurstone's [2]) do not admit the stationary-distribution interpretation we derive here.

## 4 Experimental evaluation

In this section, we compare LSR and I-LSR to other inference algorithms in terms of (*a*) statistical efficiency, and (*b*) empirical performance. In order to understand the efficiency of the estimators, we generate synthetic data from a known ground truth. Then, we look at five real-world datasets and investigate the practical performance of the algorithms in terms of accuracy, running time and convergence rate.

**Error metric.** As the probability of $i$ winning over $j$ depends on the ratio of strengths $\pi_i/\pi_i$, the strengths are typically logarithmically spaced. In order to evaluate the accuracy of an estimate $\boldsymbol{\pi}$ to ground truth parameters $\boldsymbol{\pi}^*$, we therefore use a $\log$ transformation, reminiscent of the random-utility-theoretic formulation of the choice model [1, 11]. Define $\boldsymbol{\theta} \doteq [\log \pi_i - t]$, with $t$ chosen such that $\sum_i \theta_i = 0$. We will consider the root-mean-squared error (RMSE)

$$E_{\text{RMS}} = \|\boldsymbol{\theta} - \boldsymbol{\theta}^*\|_2/\sqrt{n}. \tag{9}$$

### 4.1 Statistical efficiency

To assess the statistical efficiency of LSR and other algorithms, we follow the experimental procedure of Hajek et al. [11]. We consider $n = 1024$ items, and draw $\boldsymbol{\theta}^*$ uniformly at random in $[-2, 2]^n$. We generate $d = 64$ full rankings over the $n$ items from a Plackett-Luce model parameterized with $\boldsymbol{\pi} \propto [e^{\theta_i}]$. For a given $k \in \{2^1, \ldots, 2^{10}\}$, we break down each of the full rankings as follows. First, we partition the items into $n/k$ subsets of size $k$ uniformly at random. Then, we store the $k$-way rankings induced by the full ranking on each of those subsets. As a result, we obtain $m = dn/k$ statistically independent $k$-way partial rankings. For a given estimator, this data produces an estimate $\boldsymbol{\theta}$, for which we record the root-mean-square error to $\boldsymbol{\theta}^*$. We consider four estimators. The first two (LSR and ML) work on the ranking data directly. The remaining two follow Azari Soufiani et al. [10], who suggest breaking down $k$-way rankings into $\binom{k}{2}$ pairwise comparisons. These comparisons are then used by LSR, resulting in Azari Soufiani et al.'s GMM-F estimator, and by an ML estimator (ML-F.) In short, the four estimators vary according to (*a*) whether they use as-is rankings or derived comparisons, and (*b*) whether the model is fitted using an approximate spectral algorithm or using exact maximum likelihood. Figure 1 plots $E_{\text{RMS}}$ for increasing sizes of partial rankings, as well as a lower bound to the error of any estimator for the Plackett-Luce model (see Hajek et al. [11] for details.) We observe that breaking the rankings into pairwise comparisons (*-F estimators) incurs a significant efficiency loss over using the $k$-way rankings directly (LSR and ML.) We conclude that by correctly weighting pairwise rates in the Markov chain, LSR distinctly outperforms the rank-breaking approach as $k$ increases. We also observe that the ML estimate is always more efficient. Spectral estimators such as LSR provide a quick, asymptotically consistent estimate of parameters, but this observation justifies calling them *approximate* inference algorithms.

### 4.2 Empirical performance

We investigate the performance of various inference algorithms on five real-world datasets. The NASCAR [18] and sushi [24] datasets contain multiway partial rankings. The YouTube, GIFGIF and chess datasets[2] contain pairwise comparisons. Among those, the chess dataset is particular in that it features 45% of ties; in this case we use the extension of the Bradley–Terry model proposed by Rao and Kupper [23]. We preprocess each dataset by discarding items that are not part of the largest strongly connected component in the comparison graph. The number of items $n$, the number of rankings $m$, as well as the size of a partial rankings $k$ for each dataset are given in Table 1. Additional details on the experimental setup are given in the supplementary material. We first compare the estimates produced by three approximate ML inference algorithms, LSR, GMM-F and Rank Centrality (RC.) Note that RC applies only to pairwise comparisons, and that LSR is the only

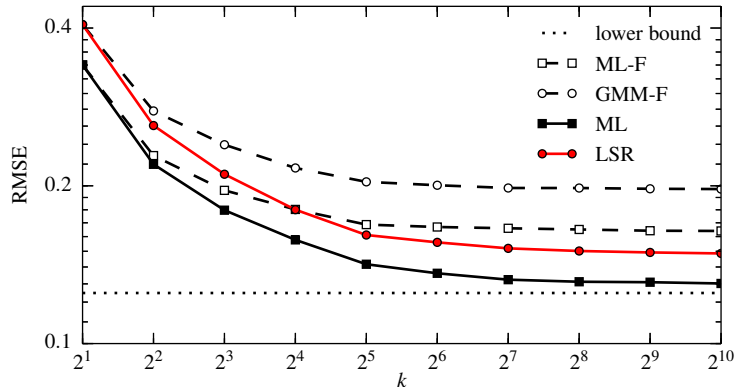

Figure 1: Statistical efficiency of different estimators for increasing sizes of partial rankings. As $k$ grows, breaking rankings into pairwise comparisons becomes increasingly inefficient. LSR remains efficient at no additional computational cost.

algorithm able to infer the parameters in the Rao-Kupper model. Also note that in the case of pairwise comparisons, GMM-F and LSR are strictly equivalent. In Table 1, we report the root-mean-square deviation to the ML estimate $\hat{\boldsymbol{\theta}}$ and the running time $T$ of the algorithm.

Table 1: Performance of approximate ML inference algorithms

| Dataset | $n$ | $m$ | $k$ | LSR $E_{\text{RMS}}$ | LSR $T$ [s] | GMM-F $E_{\text{RMS}}$ | GMM-F $T$ [s] | RC $E_{\text{RMS}}$ | RC $T$ [s] |
|---|---|---|---|---|---|---|---|---|---|
| NASCAR | 83 | 36 | 43 | **0.194** | 0.03 | 0.751 | 0.06 | — | — |
| Sushi | 100 | 5 000 | 10 | **0.034** | 0.22 | 0.130 | 0.19 | — | — |
| YouTube | 16 187 | 1 128 704 | 2 | **0.417** | 34.18 | **0.417** | 34.18 | 0.432 | 41.91 |
| GIFGIF | 5 503 | 95 281 | 2 | **1.286** | 1.90 | **1.286** | 1.90 | 1.295 | 2.84 |
| Chess | 6 174 | 63 421 | 2 | **0.420** | 2.90 | — | — | — | — |

The smallest value of $E_{\text{RMS}}$ is highlighted in bold for each dataset. We observe that in the case of multiway partial rankings, LSR is almost four times more accurate than GMM-F on the datasets considered. In the case of pairwise comparisons, RC is slightly worse than LSR and GMM-F, because the number of comparisons per pair is not homogeneous (see Section 3.3.) The running time of the three algorithms is comparable.

Next, we turn our attention to ML inference and consider three iterative algorithms: I-LSR, MM and Newton-Raphson. For Newton-Raphson, we use an off-the-shelf solver. Each algorithm is initialized with $\boldsymbol{\pi}^{(0)} = [1/n, \ldots, 1/n]^\intercal$, and convergence is declared when $E_{\text{RMS}} < 0.01$. In Table 2, we report the number of iterations $I$ needed to reach convergence, as well as the total running time $T$ of the algorithm.

Table 2: Performance of iterative ML inference algorithms.

| Dataset | $\gamma_{\mathcal{D}}$ | I-LSR $I$ | I-LSR $T$ [s] | MM $I$ | MM $T$ [s] | Newton $I$ | Newton $T$ [s] |
|---|---|---|---|---|---|---|---|
| NASCAR | 0.832 | 3 | **0.08** | 4 | 0.10 | — | — |
| Sushi | 0.890 | 2 | **0.42** | 4 | 1.09 | 3 | 10.45 |
| YouTube | 0.002 | 12 | **414.44** | 8 680 | 22 443.88 | — | — |
| GIFGIF | 0.408 | 10 | **22.31** | 119 | 109.62 | 5 | 72.38 |
| Chess | 0.007 | 15 | **43.69** | 181 | 55.61 | 3 | 49.37 |

The smallest total running time $T$ is highlighted in bold for each dataset. We observe that Newton-Raphson does not always converge, despite the log-likelihood being strictly concave[3]. I-LSR consis-

tently outperforms MM and Newton-Raphson in running time. Even if the average running time per iteration is in general larger than that of MM, it needs considerably fewer iterations: For the YouTube dataset, I-LSR yields an increase in speed of over $50$ times.

The slow convergence of minorization-maximization algorithms is known [18], yet the scale of the issue and its apparent unpredictability is surprising. In Hunter's MM algorithm, updating a given $\pi_i$ involves only parameters of items to which $i$ has been compared. Therefore, we speculate that the convergence rate of MM is dependent on the expansion properties of the comparison graph $G_{\mathcal{D}}$. As an illustration, we consider the sushi dataset. To quantify the expansion properties, we look at the spectral gap $\gamma_{\mathcal{D}}$ of a simple random walk on $G_{\mathcal{D}}$; intuitively, the larger the spectral gap is, the better the expansion properties are [12]. The original comparison graph is almost complete, and $\gamma_{\mathcal{D}} = 0.890$. By breaking each 10-way ranking into $5$ independent pairwise comparisons, we effectively sparsify the comparison graph. As a result, the spectral gap decreases to $0.784$. In Figure 2, we show the convergence rate of MM and I-LSR for the original ($k = 10$) and modified ($k = 2$) datasets. We observe that both algorithms display linear convergence, however the rate at which MM converges appears to be sensitive to the structure of the comparison graph. In contrast, I-LSR is robust to changes in the structure. The spectral gap of each dataset is listed in Table 2.

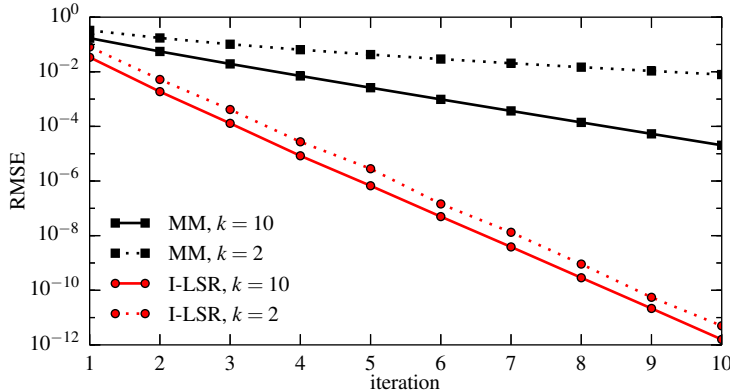

Figure 2: Convergence rate of I-LSR and MM on the sushi dataset. When partial rankings ($k = 10$) are broken down into independent comparisons ($k = 2$), the comparison graph becomes sparser. I-LSR is robust to this change, whereas the convergence rate of MM significantly decreases.

## 5  Conclusion

In this paper, we develop a stationary-distribution perspective on the maximum-likelihood estimate of Luce's choice model. This perspective explains and unifies several recent spectral algorithms from an ML inference point of view. We present our own spectral algorithm that works on a wider range of data, and show that the resulting estimate significantly outperforms previous approaches in terms of accuracy. We also show that this simple algorithm, with a straighforward adaptation, can produce a sequence of estimates that converge to the ML estimate. On real-world datasets, our ML algorithm is always faster than the state of the art, at times by up to two orders of magnitude.

Beyond statistical and computational performance, we believe that a key strength of our algorithms is that they are simple to implement. As an example, our implementation of LSR fits in ten lines of Python code. The most complex operation—finding a stationary distribution—can be readily offloaded to commonly available and highly optimized linear-algebra primitives. As such, we believe that our work is very useful for practitioners.

#### Acknowledgments

We thank Holly Cogliati-Bauereis, Ksenia Konyushkova and Brunella Spinelli for careful proofreading and comments on the text.

---

of starting point, step size, or by monitoring the numerical stability; however, these modifications are non-trivial and impose an additional burden on the practitioner.

## Footnotes

[1] This is equivalent to stating that the hypergraph $H = (V, \mathcal{A})$ is connected.

[2] See `https://archive.ics.uci.edu/ml/machine-learning-databases/00223/`, `http://www.gif.gf/` and `https://www.kaggle.com/c/chess`.

[3] On the NASCAR dataset, this has also been noted by Hunter [18]. Computing the Newton step appears to be severely ill-conditioned for many real-world datasets. We believe that it can be addressed by a careful choice

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
