[Supplementary Material]

# Fast and Accurate Inference of Plackett–Luce Models
# Supplementary Material

**Lucas Maystre**
EPFL
lucas.maystre@epfl.ch

**Matthias Grossglauser**
EPFL
matthias.grossglauser@epfl.ch

## 1   Stationary Points of the Log-Likelihood

In this section, we briefly explain why the log-likelihood in Luce's model has a unique stationary point, that at the ML estimate. Recall that we assume that the comparison graph $G_\mathcal{D}$ is strongly connected. The log-likelihood is given by

$$\log \mathcal{L}(\boldsymbol{\pi} \mid \mathcal{D}) = \sum_{\ell=1}^{d} \left( \log \pi_\ell - \log \sum_{j \in A_\ell} \pi_j \right). \tag{1}$$

This function is not concave in $\boldsymbol{\pi}$; however, this does not preclude the existence of a unique stationary point. Letting $\pi_i = e^{\theta_i}$, we write the reparametrized log-likelihood as

$$\log \mathcal{L}(\boldsymbol{\pi}(\boldsymbol{\theta}) \mid \mathcal{D}) = \sum_{\ell=1}^{d} \left( \theta_\ell - \log \sum_{j \in A_\ell} e^{\theta_j} \right),$$

which is strictly concave in $\boldsymbol{\theta}$ and therefore admits a unique stationary point, at the maximum of the function. Denote this maximum by $\hat{\boldsymbol{\theta}}$. The partial derivative of the log-likelihood with respect to $\pi_\ell$ is

$$\frac{\partial \log \mathcal{L}}{\partial \pi_\ell} = \frac{\partial \log \mathcal{L}}{\partial \theta_i} \cdot \frac{\partial \theta_i}{\partial \pi_i} = \frac{\partial \log \mathcal{L}}{\partial \theta_i} \cdot \frac{1}{\pi_i}. \tag{2}$$

As $1/\pi_i$ is strictly positive, the partial derivative vanishes only at $\hat{\pi}_i = e^{\hat{\theta}_i}$. In conclusion, $\hat{\boldsymbol{\pi}}$ is the unique ML estimate, as well as the only stationary point.

## 2   Proofs of Theorems 1 and 2

For any two items $i$ and $j$, recall that $\mathcal{D}_{i \succ j} \subseteq \mathcal{D}$ is the set of observations where $i$ wins over $j$. Let $\Delta_n = \{ \boldsymbol{u} \in \mathbf{R}^n \mid u_i > 0, \sum_i u_i = 1 \}$ be the open $(n-1)$-dimensional simplex. Recall that for $\mathcal{S} \subseteq \mathcal{D}$ and $\boldsymbol{\pi} \in \Delta_n$, we define

$$f(\mathcal{S}, \boldsymbol{\pi}) \doteq \sum_{A \in \mathcal{S}} \frac{1}{\sum_{i \in A} \pi_i}. \tag{3}$$

We will now prove the following theorem.

**Theorem 1.** *The Markov chain with inhomogeneous transition rates $\lambda_{ji} = f(\mathcal{D}_{i \succ j}, \boldsymbol{\pi})$ converges to the maximum-likelihood estimate $\hat{\boldsymbol{\pi}}$, for any initial distribution $\boldsymbol{\pi}^0 \in \Delta_n$.*

We take a discrete-time perspective, and consider the uniformized Markov chain with (parametric) transition probabilities

$$P(\boldsymbol{\pi})_{ij} = \begin{cases} \epsilon \displaystyle\sum_{A \in \mathcal{D}_{j \succ i}} \frac{1}{\sum_{t \in A} \pi_t} & \text{if } j \neq i, \\[2em] 1 - \epsilon \displaystyle\sum_{k \neq i} \sum_{A \in \mathcal{D}_{k \succ i}} \frac{1}{\sum_{t \in A} \pi_t} & \text{if } j = i, \end{cases} \tag{4}$$

where $\epsilon$ (the uniform rate parameter) is a small factor that ensures that the matrix is row-stochastic. We say that the Markov chain is *inhomogeneous* because the transition probabilities depend on the current distribution over states; as a consequence, standard ergodic results do not apply directly. From the development at the beginning of Section 3 of the main text, it follows that $\hat{\boldsymbol{\pi}}$ is the unique invariant distribution of the Markov chain, i.e., satisfying $\hat{\boldsymbol{\pi}} = \hat{\boldsymbol{\pi}} P(\hat{\boldsymbol{\pi}})$. Consider the mapping $T : \Delta_n \to \Delta_n$ defined by

$$T(\boldsymbol{\pi}) = \boldsymbol{\pi} P(\boldsymbol{\pi}), \tag{5}$$

representing the distribution after one step of the Markov chain. Using a contraction argument, we will show that the iteration $\boldsymbol{\pi}^{k+1} = T(\boldsymbol{\pi}^k)$ converges to a fixed point for any $\boldsymbol{\pi}^0 \in \Delta_n$. It directly follows that the Markov chain converges to $\hat{\boldsymbol{\pi}}$ from any initial distribution.

We start with a technical lemma that characterizes the Jacobian matrix of the mapping. We will use the notation

$$T^k(\boldsymbol{\pi}) = \underbrace{T \circ T \circ \ldots \circ T}_{k \text{ times}}(\boldsymbol{\pi}) \tag{6}$$

for $k$ successive applications of the mapping. We will also extend our notation for subsets of observations, and let $\mathcal{D}_{i \succ j,k} \subseteq \mathcal{D}$ be the observations where $i$ wins among a set of alternatives containing $j$ and $k$.

**Lemma 1.** *The Jacobian matrix of the mapping $T(\boldsymbol{\pi})$ defined in (5) is given by*

$$T'(\boldsymbol{\pi})_{ij} = \left[ \frac{\partial T(\boldsymbol{\pi})}{\partial \pi_i} \right]_j = \begin{cases} \epsilon \sum_k \sum_{A \in \mathcal{D}_{k \succ j,i}} \dfrac{\pi_j}{(\sum_{t \in A} \pi_t)^2} & \text{if } j \neq i, \\[3mm] 1 - \epsilon \sum_{j \neq \ell} \sum_k \sum_{A \in \mathcal{D}_{k \succ j,\ell}} \dfrac{\pi_j}{(\sum_{t \in A} \pi_t)^2} & \text{if } j = i. \end{cases} \tag{7}$$

*Furthermore, there is a finite $m \in \mathbf{N}$ such that for $S' = (T^m)'$ it holds that $\delta = \min_{i,j} S'_{ij} > 0$ and $\|S'\|_1 = 1$.*

*Proof.* The partial derivative of $T$ with respect to $\pi_\ell$ at $j \neq \ell$ is

$$\left[ \frac{\partial T(\boldsymbol{\pi})}{\partial \pi_\ell} \right]_j = \left[ \frac{\partial \boldsymbol{\pi}}{\partial \pi_\ell} P(\boldsymbol{\pi}) \right]_j + \left[ \boldsymbol{\pi} \frac{\partial P(\boldsymbol{\pi})}{\partial \pi_\ell} \right]_j \tag{8}$$

$$= \epsilon \sum_{A \in \mathcal{D}_{j \succ \ell}} \frac{1}{\sum_{t \in A} \pi_t} - \epsilon \sum_{k \neq j} \sum_{A \in \mathcal{D}_{j \succ k,\ell}} \frac{\pi_k}{(\sum_{t \in A} \pi_t)^2}$$

$$\qquad + \epsilon \sum_{k \neq j} \sum_{A \in \mathcal{D}_{k \succ j,\ell}} \frac{\pi_j}{(\sum_{t \in A} \pi_t)^2} \tag{9}$$

$$= \epsilon \sum_{A \in \mathcal{D}_{j \succ \ell}} \frac{\pi_j}{(\sum_{t \in A} \pi_t)^2} + \epsilon \sum_{k \neq j} \sum_{A \in \mathcal{D}_{k \succ j,\ell}} \frac{\pi_j}{(\sum_{t \in A} \pi_t)^2} \tag{10}$$

$$= \epsilon \sum_k \sum_{A \in \mathcal{D}_{k \succ j,\ell}} \frac{\pi_j}{(\sum_{t \in A} \pi_t)^2}. \tag{11}$$

To go from (9) to (10), we reverse the order of summation in the subtracted term and rewrite the fraction inside the left term.

$$\sum_{A \in \mathcal{D}_{j \succ \ell}} \frac{1}{\sum_{t \in A} \pi_t} - \sum_{k \neq j} \sum_{A \in \mathcal{D}_{j \succ k,\ell}} \frac{\pi_k}{(\sum_{t \in A} \pi_t)^2} \tag{12}$$

$$= \sum_{A \in \mathcal{D}_{j \succ \ell}} \frac{1}{\sum_{t \in A} \pi_t} - \sum_{A \in \mathcal{D}_{j \succ \ell}} \sum_{k \in A, k \neq j} \frac{\pi_k}{(\sum_{t \in A} \pi_t)^2} \tag{13}$$

$$= \sum_{A \in \mathcal{D}_{j \succ \ell}} \sum_{k \in A} \frac{\pi_k}{(\sum_{t \in A} \pi_t)^2} - \sum_{A \in \mathcal{D}_{j \succ \ell}} \sum_{k \in A, k \neq j} \frac{\pi_k}{(\sum_{t \in A} \pi_t)^2} \tag{14}$$

$$= \sum_{A \in \mathcal{D}_{j \succ \ell}} \frac{\pi_j}{(\sum_{t \in A} \pi_t)^2} \tag{15}$$

One can find the partial derivative with respect to $\pi_\ell$ at $\ell$ by noticing that each row of the Jacobian matrix sums to one:

$$\sum_j \left[ \frac{\partial T(\boldsymbol{\pi})}{\partial \pi_\ell} \right]_j = \sum_j P(\boldsymbol{\pi})_{\ell j} + \sum_j \sum_i \pi_i \frac{\partial P(\boldsymbol{\pi})_{ij}}{\partial \pi_\ell} \tag{16}$$

$$= 1 + \sum_i \pi_i \frac{\partial}{\partial \pi_\ell} \sum_j P(\boldsymbol{\pi})_{ij} = 1. \tag{17}$$

The matrix is therefore row-stochastic, and $\|T'(\boldsymbol{\pi})\|_1 = 1$. Because transition probabilities are strictly positive on the edges of the comparison graph (which is, by assumption, strongly connected), there is a finite $m \in \mathbf{N}$ such that all entries of $T^m(\boldsymbol{\pi})$ are lower-bounded by a strictly positive number. It is easy to see that the Jacobian matrix $T'$ also has strictly positive entries on the edges of the comparison graph, and therefore

$$S'(\boldsymbol{\pi}) = (T^m(\boldsymbol{\pi}))' = \prod_{i=0}^{m-1} T'(T^i(\boldsymbol{\pi})) \tag{18}$$

also has its entries lower-bounded by a strictly positive number. Furthermore, $S'(\boldsymbol{\pi})$ is a product of stochastic matrices, hence $\|S'(\boldsymbol{\pi})\|_1 = 1$. $\qquad\square$

Now we will use the properties of the Jacobian matrix to show that $T$ is a fixed-point iteration, using a standard argument. Our proof is inspired by the lecture notes of Tresch [1] and von Petersdorff [2].

*Proof of Theorem 1.* Using the results of Lemma 1, let $S(\boldsymbol{\pi}) = T^m(\boldsymbol{\pi})$ and write $S'(\boldsymbol{\pi})$ as

$$S'(\boldsymbol{\pi}) = \delta 1_{n \times n} + R(\boldsymbol{\pi}), \tag{19}$$

where $1_{n \times n}$ is the all-ones matrix, and $\|R(\boldsymbol{\pi})\|_1 = 1 - n\delta = c < 1$. Now pick any $\boldsymbol{x}, \boldsymbol{y} \in \Delta_n$, and let $\tilde{S}(u) \doteq S(\boldsymbol{x} + u(\boldsymbol{x} - \boldsymbol{y}))$. Then $\tilde{S}'(u) = S'(\boldsymbol{x} + u(\boldsymbol{y} - \boldsymbol{x}))(\boldsymbol{y} - \boldsymbol{x})$, and

$$S(\boldsymbol{y}) - S(\boldsymbol{x}) = \tilde{S}(1) - \tilde{S}(0) = \int_0^1 \tilde{S}'(u) du = \int_0^1 S'(\boldsymbol{x} + u(\boldsymbol{y} - \boldsymbol{x}))(\boldsymbol{y} - \boldsymbol{x}) du \tag{20}$$

As $S'$ is continuous, we have

$$\|S(\boldsymbol{y}) - S(\boldsymbol{x})\|_1 \leq \int_0^1 \|S'(\boldsymbol{x} + u(\boldsymbol{y} - \boldsymbol{x}))(\boldsymbol{y} - \boldsymbol{x})\|_1 du \tag{21}$$

$$= \int_0^1 \| \underbrace{\delta 1_{n \times n}(\boldsymbol{y} - \boldsymbol{x})}_{=0} + R(\boldsymbol{x} + u(\boldsymbol{y} - \boldsymbol{x}))(\boldsymbol{y} - \boldsymbol{x}) \|_1 du \tag{22}$$

$$\leq \int_0^1 \underbrace{\|R(\boldsymbol{x} + u(\boldsymbol{y} - \boldsymbol{x}))\|_2}_{\leq c} \|\boldsymbol{y} - \boldsymbol{x}\|_1 du \tag{23}$$

$$\leq c \|\boldsymbol{y} - \boldsymbol{x}\|_1 \tag{24}$$

Therefore, by the contraction mapping principle, the sequence of iterates $\boldsymbol{\pi}^{k+1} = T^m(\boldsymbol{\pi}^k)$ converges to $\hat{\boldsymbol{\pi}}$. Finally, we observe that for any $\boldsymbol{\pi} \in \Delta_n$, the vectors $\boldsymbol{\pi}, T(\boldsymbol{\pi}), T^2(\boldsymbol{\pi}), \dots$ occur in one of the sequences

$$\left( S^k(T^r(\boldsymbol{\pi})) \right)_{k \in \mathbf{N}_0}, \quad r \in \{0, \dots, m-1\}. \tag{25}$$

All sequences converge to $\hat{\boldsymbol{\pi}}$, and therefore

$$\lim_{k \to \infty} T^k(\boldsymbol{\pi}) = \hat{\boldsymbol{\pi}}. \tag{26}$$

$\qquad\square$

**Theorem 2.** *Let $\mathcal{A} = \{A_\ell\}$ be a collection of sets of alternatives such that for any partition of $\mathcal{A}$ into two non-empty sets $S$ and $T$, $(\cup_{A \in S} A) \cap (\cup_{A \in T} A) \neq \varnothing$. Let $d_\ell$ be the number of choices observed over alternatives $A_\ell$. Then $\bar{\boldsymbol{\pi}} \to \boldsymbol{\pi}^*$ as $d_\ell \to \infty \; \forall \ell$.*

*Proof.* Let $d \to \infty$ be a shorthand for $d_\ell \to \infty \ \forall \ell$. The condition on $\mathcal{A}$ is equivalent to stating that the hypergraph $H = (V, \mathcal{A})$ with $V = \{1, \ldots, n\}$ is connected. First, we show that asymptotically, the graph $G_\mathcal{D} = (V, E)$ is connected. For a given set of alternatives $A_\ell$, let $i, j \in A_\ell$. The probability that $(j, i) \in E$ is

$$1 - \left(1 - \frac{\pi_i}{\sum_{t \in A_\ell} \pi_t}\right)^{d_\ell} > 1 - (1 - \pi_i)^{d_\ell} \xrightarrow{d_\ell \to \infty} 1, \tag{27}$$

where we use the fact that $\pi_i > 0 \ \forall i$. Therefore, asymptotically, every alternative set $A_\ell$ forms a clique in $G_\mathcal{D}$. By assumption of connectivity on the hypergraph $H$, $G_\mathcal{D}$ is strongly connected.

Now that we know that the Markov chain is asymptotically ergodic, we will show that the stationary distribution matches the true model parameters. Let $C_\ell^s$ be a random variable denoting the item chosen in the $s$-th observation over alternatives $A_\ell$. By the law of large numbers, for any item $i \in A_\ell$

$$\lim_{d_\ell \to \infty} \frac{1}{d_\ell} \sum_{s=1}^{d_\ell} \mathbf{1}\{C_\ell^s = i\} = \frac{\pi_i^*}{\sum_{t \in A_\ell} \pi_t^*}. \tag{28}$$

Now consider two items $i$ and $j$. If they have never been compared, $\lambda_{ij} = \lambda_{ji} = 0$. Otherwise, suppose that they have been compared in alternative sets whose indices are in $B = \{\ell \mid i, j \in A_\ell\}$. Let $\mathbf{1}\{X\}$ be the indicator variable for event $X$. By construction of the transition rates in LSR, we have that

$$\frac{\lambda_{ij}}{\lambda_{ji}} = \frac{\sum_{\ell \in B} \sum_{s=1}^{d_\ell} \mathbf{1}\{C_\ell^s = j\} \, n/|A_\ell|}{\sum_{\ell \in B} \sum_{s=1}^{d_\ell} \mathbf{1}\{C_\ell^s = i\} \, n/|A_\ell|}. \tag{29}$$

From (28) it follows that

$$\lim_{d \to \infty} \frac{\lambda_{ij}}{\lambda_{ji}} = \frac{\sum_{\ell \in B} (\pi_j^* / \sum_{t \in A_\ell} \pi_t^*) \, n/|A_\ell|}{\sum_{\ell \in B} (\pi_i^* / \sum_{t \in A_\ell} \pi_t^*) \, n/|A_\ell|} \tag{30}$$

$$= \frac{\pi_j^*}{\pi_i^*} \cdot \frac{\sum_{\ell \in B} (1/\sum_{t \in A_\ell} \pi_t^*) \, n/|A_\ell|}{\sum_{\ell \in B} (1/\sum_{t \in A_\ell} \pi_t^*) \, n/|A_\ell|} = \frac{\pi_j^*}{\pi_i^*}. \tag{31}$$

Therefore, when $d \to \infty$,

$$\sum_{j \neq i} \pi_i^* \lambda_{ij} = \sum_{j \neq i} \pi_i^* \left(\frac{\pi_j^*}{\pi_i^*} \lambda_{ji}\right) = \sum_{j \neq i} \pi_j^* \lambda_{ji} \quad \forall i. \tag{32}$$

It is easy to recognize the global balance equations, and it follows that $\boldsymbol{\pi}^*$ is the stationary distribution of the asymptotical Markov chain. $\qquad \square$

## 3 Bound on error rate of ML estimate

We use the analytical framework of Negahban et al. [3] to bound the error rate of the ML estimator in the case where (*a*) the data is in the form of pairwise comparisons and (*b*) for each pair under comparison, we observe exactly $k$ outcomes.

Let $G = (V, E)$ be an undirected graph where $V = \{1, \ldots, n\}$ and $(i, j) \in E$ if $i$ and $j$ have been compared. Let $d_{\min}$ and $d_{\max}$ be the minimum and maximum degree of a node in $G$, respectively. Let $\gamma$ be the spectral gap of a simple random walk on $G$; intuitively, the larger the spectral gap is, the faster the convergence to the stationary distribution is. For each $(i, j) \in E$ we observe $k$ comparisons generated from ground truth parameters $\boldsymbol{\pi}^*$. Let $A_{ji}$ denote the number of times $i$ wins against $j$ and $a_{ji} = A_{ji}/k$ the ratio of wins of $i$ over $j$. We say that an event $X$ occurs with high probability if $\mathbf{P}(X) \geq 1 - c/n^\alpha$ for $c, \alpha$ fixed.

**Theorem 3.** *For $k \geq 4C^2(1 + (b^6 \kappa^2/(d_{\max} \gamma^2))) \log n$, the error on the ML estimate $\hat{\boldsymbol{\pi}}$ satisfies w.h.p.*

$$\frac{\|\hat{\boldsymbol{\pi}} - \boldsymbol{\pi}^*\|_2}{\|\boldsymbol{\pi}^*\|_2} < C \frac{b^{7/2} \kappa}{\gamma} \sqrt{\frac{\log n}{k d_{\max}}}, \tag{33}$$

*where $C$ is a constant, $b = \max_{i,j} \pi_i^*/\pi_j^*$ and $\kappa = d_{\max}/d_{\min}$.*

*Proof.* The ML estimate can be interpreted as the stationary distribution of the discrete-time Markov chain

$$\widehat{P}_{ij} = \begin{cases} \epsilon \dfrac{a_{ij}}{\hat{\pi}_i + \hat{\pi}_j} & \text{if } i \neq j, \\ 1 - \epsilon \displaystyle\sum_{l \neq i} \dfrac{a_{il}}{\hat{\pi}_i + \hat{\pi}_l} & \text{if } i = j. \end{cases} \tag{34}$$

The factor $\epsilon = \hat{\pi}_{\min}/d_{\max}$ ensures that $\widehat{P}$ is stochastic. Given this matrix, it is straightforward to analyze the $\hat{\boldsymbol{\pi}}$ by using the methods developed for Rank Centrality (RC); the proof essentially follows that of Theorem 1 of Negahban et al. [3]. Let $P^*$ be the ideal Markov chain, when $a_{ij} = \pi_j^*/(\pi_i^* + \pi_j^*)$, i.e., the ratios are noiseless. The key observation is to note that the stationary distribution of $P^*$ is $\boldsymbol{\pi}^*$, the true model parameters. By bounding $\|\widehat{P} - P^*\|_2$ and $1 - \lambda_{\max}(P^*)$, we can bound the error on the stationary distribution of $P^*$. For the former, a straightforward application of the proof in the RC case suffices. For the latter, in the application of the comparison theorem, the lower bound on $\min_{i,j} \pi_i^* P_{ij}^*$ changes by a factor of $1/(2b)$. This is due to the additional factor $\hat{\pi}_{\min}/(\hat{\pi}_i + \hat{\pi}_j)$ in the off-diagonal entries of $P^*$. $\qquad\square$

If the graph of comparisons $G$ is an expander, then $\gamma = O(1)$. Furthermore, if $d_{\max} \propto d_{\min}$, then $\kappa = O(1)$. A realization of the $G(n,p)$ random graph satisfies these two constraints with high probability as long as $p = \omega(\log n/n)$. It follows that if $\omega(n \log n)$ comparison pairs are chosen uniformly at random and $k = O(1)$ outcomes are observed for each pair, the error goes to zero as $n$ increases.

Hajek et al. [4] recently proved a more general version of our result, using a different analytical technique. Their bound is qualitatively similar, but also applies to multiway rankings and heterogeneous number of comparisons.

## 4 Derivation for the Rao–Kupper model

We consider a model that was proposed by Rao and Kupper in 1967 [5]. This model extends the Bradley–Terry model in that a comparison between two items can result in a tie. Letting $\alpha \in [1, \infty)$, the probabilities of $i$ winning over and tying with $j$, respectively, are given as follows.

$$p(i \succ j) = \frac{\pi_i}{\pi_i + \alpha \pi_j},$$

$$p(i \leftrightarrow j) = \frac{\pi_i \pi_j (\alpha^2 - 1)}{(\pi_i + \alpha \pi_j)(\alpha \pi_i + \pi_j)}.$$

This model is useful for e.g., chess, where a significant fraction of comparison outcomes do not result in either a win or a loss.

We assume that the parameter $\alpha$ is fixed, and derive an expression of the ML estimate $\hat{\boldsymbol{\pi}}$. Let $A_{ji}$ be the number of times $i$ wins over $j$, and $T_{ij} = T_{ji}$ be the number of ties between $i$ and $j$. The log-likelihood can be written as

$$\log \mathcal{L} = \sum_i \sum_{j \neq i} A_{ji} \left( \log(\pi_i) - \log(\pi_i + \alpha \pi_j) \right) \tag{35}$$

$$+ \sum_i \sum_{j > i} T_{ij} (\log(\pi_i) + \log(\pi_j) + \log(\alpha^2 - 1)$$

$$- \log(\pi_i + \alpha \pi_j) - \log(\alpha \pi_i + \pi_j)).$$

The log-likelihood function is strictly concave and the model admits a unique ML estimate $\hat{\boldsymbol{\pi}}$. The optimality condition $\nabla_{\hat{\boldsymbol{\pi}}} \log \mathcal{L} = 0$ implies

$$\frac{\partial \log \mathcal{L}}{\partial \hat{\pi}_i} = \sum_{j \neq i} A_{ji} \left( \frac{1}{\hat{\pi}_i} - \frac{1}{\hat{\pi}_i + \alpha \hat{\pi}_j} \right) - A_{ij} \frac{\alpha}{\alpha \hat{\pi}_i + \hat{\pi}_j} \tag{36}$$

$$+ T_{ij} \left( \frac{1}{\hat{\pi}_i} - \frac{1}{\hat{\pi}_i + \alpha \hat{\pi}_j} - \frac{\alpha}{\alpha \hat{\pi}_i + \hat{\pi}_j} \right) = 0 \tag{37}$$

$$\iff \sum_{j \neq i} A_{ji} \frac{\alpha \hat{\pi}_j}{\hat{\pi}_i + \alpha \hat{\pi}_j} - A_{ij} \frac{\alpha \hat{\pi}_i}{\alpha \hat{\pi}_i + \hat{\pi}_j} \tag{38}$$

$$+ T_{ij} \frac{\alpha \hat{\pi}_j^2 - \alpha \hat{\pi}_i^2}{(\hat{\pi}_i + \alpha \hat{\pi}_j)(\alpha \hat{\pi}_i + \hat{\pi}_j)} = 0 \tag{39}$$

$$\iff \sum_{j \neq i} \frac{A_{ji} + T_{ji} \frac{\hat{\pi}_j}{\alpha \hat{\pi}_i + \hat{\pi}_j}}{\hat{\pi}_i + \alpha \hat{\pi}_j} \hat{\pi}_j - \frac{A_{ij} + T_{ij} \frac{\hat{\pi}_i}{\hat{\pi}_i + \alpha \hat{\pi}_j}}{\alpha \hat{\pi}_i + \hat{\pi}_j} \hat{\pi}_i = 0. \tag{40}$$

Therefore, the ML estimate is the stationary distribution of a Markov chain with transition rates

$$\lambda_{ij} = \frac{A_{ij} + T_{ij} \frac{\hat{\pi}_i}{\hat{\pi}_i + \alpha \hat{\pi}_j}}{\alpha \hat{\pi}_i + \hat{\pi}_j}. \tag{41}$$

The extension of LSR and I-LSR to the Rao–Kupper model given these transition rates is straightforward.

## 5 Finding the stationary distribution

A set of transition rates $[\lambda_{ij}]$ that satisfy the strong connectivity assumption yields a unique stationary distribution $\boldsymbol{\pi}$. In practice, finding this stationary distribution can be implemented in various ways. We distinguish implementations based on whether they consider a continuous-time or a discrete-time perspective on Markov chains.

**Continuous-time perspective.** We consider the infinitesimal generator matrix $Q$, where $Q_{ij} \doteq \lambda_{ij}$ and $Q_{ii} \doteq -\sum_j \lambda_{ij}$. The stationary distribution satisfies $\boldsymbol{\pi} Q = 0$; this is essentially a matrix formulation of the global balance equations. Therefore, one approach to finding the steady-state distribution is to compute the rank-1 left nullspace of $Q$. This can be done e.g., by LU decomposition, a basic linear-algebra primitive. In the dense case, the running time of a typical implementation is $O(n^3)$, but highly optimized parallel implementations such as that provided by LAPACK [6] are commonly available. In the sparse case, LU decomposition can be done significantly faster using adapted algorithms, such as that of Demmel et al. [7].

**Discrete-time perspective.** Let $\epsilon < 1/\max_i |Q_{ii}|$, then $P = I + \epsilon Q$ is the transition matrix of a discrete-time Markov chain that satisfies $\boldsymbol{\pi} P = \boldsymbol{\pi}$. In this case, finding the steady-state distribution is equivalent to finding the left eigenvector associated to the leading eigenvalue of the transition matrix $P$. This is also a well-studied linear algebra problem for which plenty of efficient, off-the-shelf algorithms exist. For example, power iteration methods can find the eigenvector in a few (sparse) matrix multiplications. Beyond these well-known algorithms, the recently proposed randomized approach of Halko et al. [8] enables us to scale to truly large problem sizes ($n$ is $O(10^6)$ or more.)

For our experiments, we have implemented LSR and I-LSR using a dense LU factorization of the generator matrix. The Python code, which relies on the `numpy` and `scipy` libraries[1], is displayed in Figure 5

## 6 Experimental procedure

We give a few additional details on the procedure that we followed for the experiments of Section 4 in the main paper. All experiments were run on a machine with a quad-core 2.0 GHz Haswell processor,

```
1  import numpy as np
2  import scipy.linalg as spl
3
4  def weighted_lsr(n, rankings, weights):
5      chain = np.zeros((n, n), dtype=float)
6      for ranking in rankings:
7          sum_weights = sum(weights[x] for x in ranking)
8          for i, winner in enumerate(ranking):
9              val = 1.0 / sum_weights
10             for loser in ranking[i+1:]:
11                 chain[loser, winner] += val
12             sum_weights -= weights[winner]
13     chain -= np.diag(chain.sum(axis=1))
14     return statdist(chain)
15
16 def statdist(chain):
17     lu, piv = spl.lu_factor(generator.T)
18     res = spl.solve_triangular(lu[:-1,:-1], -lu[:-1,-1])
19     res = np.append(res, 1.0)
20     return res / res.sum()
```

Figure 1: Python implementation of one iteration of I-LSR.

and 16GB of RAM, running Mac OS X 10.9. For LSR and I-LSR, we used a slightly adapted version the code presented in Figure 5. We implemented the Rank Centrality (RC), GMM-F [9], and MM [10] algorithms in Python. For Newton-Raphson, we implemented our choice model on top of the popular statsmodels Python library[2] that provides a Newton-Raphson solver. For completeness, the Python source code containing all the functions we used is provided as a separate file in the supplementary material. We have compared our implementation of the MM algorithm to that of Hunter written in Matlab[3], and observed that ours has comparable running time.

For the chess dataset, we use the Rao–Kupper model and set the parameter $\alpha = \sqrt{2}$. Note that this parameter could also be estimated from the data, however in our experiments we focus on the performance of algorithms for estimating $\hat{\pi}$.

## Footnotes

[1] See: http://www.scipy.org/.

[2] See: http://statsmodels.sourceforge.net/

[3] See: http://sites.stat.psu.edu/~dhunter/code/btmatlab/

[10] D. R. Hunter. MM algorithms for generalized Bradley–Terry models. *The Annals of Statistics*, 32(1): 384–406, 2004.