[Reviews · NeurIPS 2015]

Submitted by Assigned_Reviewer_1

Via a connection to Markov Chains, the paper exploits spectral techniques for learning Placket-Luce models efficiently and with lower error than existing methods.

The summary mostly says it all.

To the best of my knowledge the algorithmic contributions of the paper are novel as well as the theorem proving convergence.

The results are also practically useful when compared to existing work and the paper is well-situated w.r.t. general related work and trends.

Perhaps the icing on the take then is that the algorithm is straightforward to implement.

For this reason, I believe that other researchers and practitioners will find strong reasons to use this work and build on it.

I have two (minor) questions:

(1) One question relates to the error metric above Eq (9): I don't fully understand the definition of the theta vector... it appears that a scalar is assigned to a vector... I am intepreting the notation as the assignment to the i'th element of theta.

What is most confusing though is the purpose of t... why is it necessary to "center" the theta_i purpose of error comparison?

(2) The authors state in Figure 2 that I-LSR is robust to changes in graph structure.

While I agree that the impact of graph structure is more prominent for MM, I-LSR's RMSE is not immune to changes in graph structure... am I missing something?

And one simple comment:

Because the implementation of the stationary distribution computation is critical to the efficiency of Algorithms 1 and 2, some discussion of this should ideally be moved from the Appendix and into the main body of the paper (perhaps as a footnote to save space).
Summary: A solid paper of both theoretical and practical importance for the preference learning community.

Submitted by Assigned_Reviewer_2

This paper interprets ML estimation in the classic Packett-Luce models for choice, as the problem of finding the stationary distribution of a Markov chain built from the data. It then provides two natural algorithms to compute these, based on algorithms to compute stationary distributions.

Overall, the connection to markov chains is not a new one (has appeared before, at least for pairwise comparisons). The experiments are interesting. The analysis is not hard or complicated. So, maybe an interesting observation but somewhat short of the level of contribution of a full paper.
Summary: Authors observe a connection between ML inference in a classic choice model, and computing stationary distrbution of a markov chain. They use this to provide two natural algorithms for the ML problem. Overall, an interesting observation but not a very big contribution.

Submitted by Assigned_Reviewer_3

This paper propose a new inference mechanism for the Plackett-Luce model based on the preliminary observation that the ML estimate can be seen as the stationary distribution of a certain Markov chain. In fact, two inferences mechanisms are proposed, one is approximate and consistent, the other converges to the ML estimate but is slower. The authors then debate on the application settings (pairwise preferences, partial rankings). Finally, the authors exhibit three sets of experiments. The first one compares the proposed algorithm to other approximate inference mechanisms for the PL model in terms of statistical efficiency. Then on real-world datasets, one experiment compares the empirical performance of the approximate methods and a second the speed of exact methods to reach a certain level of optimality.

I found the paper extremely clear and easy to follow. I especially liked the effort of the authors to propose sketches of proof. Moreover, the related work is correctly integrated and looks exhaustive.

The problem considered (inference of PL) is not new and there already exist solutions. However, the method proposed here has the advantages to be more simple and efficient than existing ones.
Summary: A well-written paper that propose a new, simple and efficient method for the inference of Plackett-Luce model.

Submitted by Assigned_Reviewer_4

The Bradley-Terry model choice model and the related Plackett-Luce ranking model are considered in this paper. It is shown that the maximum likelihood estimate of these models can be expressed as a stationary distribution of a Markov chain. This fact is utilized to give insights into spectral inference algorithms for these models as well as to formulate a new spectral approximate inference algorithm that is more accurate than existing alternatives.

Empirical results show that the proposed approach has better statistical efficiency than alternative approximate maximum likelihood methods. Does the proposed approach have any known disadvantages compared to alternatives (e.g. model assumptions limiting applicability)?

Summary: A new spectral approximate inference algorithm that is more accurate than existing alternatives is proposed for Bradley-Terry and Plackett-Luce models.

Author Feedback
Author rebuttal: We thank all the reviewers for their insightful comments, which will be very useful in clarifying our thoughts and improving our manuscript. Below, we respond to the specific questions and issues that were raised by reviewers 1, 5, and 6; as the other reviewers did not raise specific concerns that we can address in this rebuttal, we acknowledge their valuable feedback without further comment.

# Reviewer 1

(1) Regarding the notation used in the definition above Eq. 9, when defining $\vec{\theta}$: We use the square brackets to indicate that the left-hand side is a vector, with value $\log \pi_i - t$ at the i-th index. The notation is (somewhat implicitly) introduced on the third line of the paragraph "Discrete choice model", on page 2, when defining the vector $\vec{\pi}$.

You are correct in pointing out that centering the theta parameters is not absolutely necessary. Centering $\vec{\theta}$ and $\vec{\theta}^*$ has the effect of centering the difference vector in Eq. 9; this can be seen as removing the "DC component" when computing the RMSE, and is justified by the fact that theta parameters are defined only up to an additive constant. Nevertheless we ran the experiments again without centering, i.e. using $\theta_i = \log pi_i$. We observed no qualitative difference, and only minor quantitative variations in the error. For example, for the first two rows of Table 1, the values without centering would be:

NASCAR, LSR: E_RMS = 0.197
NASCAR, GMM-F: E_RMS = 0.840
Sushi, LSR: E_RMS = 0.034
Sushi, GMM-F E_RMS 0.140

(2) Regarding the robustness of I-LSR and MM to changes in the graph structure (Figure 2): you are correct in pointing out that I-LSR takes slightly longer in order to reach a desired level of accuracy in the sparse case (dashed line) in comparison to the dense case (solid line.) This is because after the first iteration, the estimate is a little bit worse. However, we believe that what really matters is the slope: In the case of I-LSR, the curve's slope (the rate of convergence) appears to be unaffected by the sparsification---as opposed to MM, where the curve becomes much flatter. In that sense, the convergence rate is robust to changes to the graph's structure.

To further extend on this: we also experimented with synthetic graphs. Using graphs that have very poor mixing properties (e.g. a very long cycle graph, or a regular ring lattice), we can make the error slope for MM arbitrarily flat, making it virtually impossible to attain any desired level of accuracy in a reasonable time. For I-LSR, the ordinate at 1 (i.e. the residual error after the first iteration) can increase a little, but the rate of convergence stays constant for these slow-mixing graphs.

# Reviewer 5

Regarding known disadvantages: to the best of our knowledge, our algorithms do not have any particular limitation when compared to competing algorithms for the Plackett-Luce family of models. In particular (and in contrast to some competing algorithms), we support many model variants (comparisons, rankings, ties, and possibly more.)

However, this entire family of models is based on the assumption that the odds of choosing an item over another are independent of the rest of the alternatives (Eq. 1). This assumption is incorrect in several situations of practical interest, in which case more flexible choice models might be warranted. In these cases, our techniques will likely not be applicable. Nevertheless the Plackett-Luce family of models remains widely used across many domains, perhaps because of their simplicity, and we believe that improving inference in these models is highly relevant.

# Reviewer 6

Regarding the originality of the connection to Markov chains: two relatively recent NIPS papers cited in our introduction indeed observe (in one case implicitly) a connection between the stationary distribution of a Markov chain and a model from the Plackett-Luce family. The salient, original contributions of our work are:

- the mathematical derivation of a link that explains and unifies the previous observations. This insight is original, and besides unifying previous work it also translates to useful new algorithms,
- a spectral algorithm that yields a major increase in statistical efficiency with respect to the previous state of the art, at no additional computational cost. On the two k-way ranking datasets we use, we observe a 4x improvement.
- a new exact ML inference algorithm that is faster and less sensitive to the structure of the comparison graph.

In summary, we think that advances are important in the inference of models in the PL family because of their broad practical and theoretical relevance.